# Causal Mean Field Multi-Agent Reinforcement Learning

## Abstract

Scalability remains a challenge in multi-agent reinforcement learning and is currently under active research. A framework named mean-field reinforcement learning (MFRL) could alleviate the scalability problem by employing Mean Field Theory to turn a many-agent problem into a two-agent problem. However, this framework lacks the ability to identify essential interactions under non-stationary environments. Causality contains relatively invariant mechanisms behind interactions, though environments are non-stationary. Therefore, we propose an algorithm called causal mean-field Q-learning (CMFQ) to address the scalability problem. CMFQ is ever more robust toward the change of the number of agents though inheriting the compressed representation of MFRL's action-state space. Firstly, we model the causality behind the decision-making process of MFRL into a structural causal model (SCM). Then the essential degree of each interaction is quantified via intervening on the SCM. Furthermore, we design the causality-aware compact representation for behavioral information of agents as the weighted sum of all behavioral information according to their causal effects. We test CMFQ in a mixed cooperative-competitive game and a cooperative game. The result shows that our method has excellent scalability performance in both training in environments containing a large number of agents and testing in environments containing much more agents.

## 1 Introduction

multi-agent reinforcement learning (MARL) has achieved remarkable success in some challenging tasks. e.g., video games(Vinyals et al., 2019; Wu, 2019). However, training a large number of agents remains a challenge in MARL. The main reasons are 1) the dimensionality of joint state-action space increases exponentially as agent number increases, and 2) during the training for a single agent, the policies of other agents keep changing, causing the non-stationarity problem, whose severity increases as agent number increases.(Sycara, 1998; Zhang et al., 2019; Gronauer & Diepold, 2021). Existing works generally use the centralized training and decentralized execution paradigm to mitigate the scalability problem via mitigating the non-stationarity problem(Rashid et al., 2018; Foerster et al., 2018; Lowe et al., 2017; Sunehag et al., 2017). Curriculum learning and attention techniques are also used to improve the scalability performance(Long et al., 2020; Iqbal & Sha, 2019). However, above methods focus mostly on tens of agents. For large-scale multi-agent system (MAS) contains hundreds of agents, studies in game theory(Blume, 1993) and mean-field theory(Stanley, 1971; Yang et al., 2018) offers a feasible framework to mitigate the scalability problem. Under this framework, Yang et al. (2018) propose a algorithm called mean-field Q-learning (MFQ), which replaces joint action in joint Q-function with average action, assuming that the entire agent-wise interactions could be simplified into the mean of local pairwise interactions. That is, MFQ reduces the dimensionality of joint state-action space with a merged agent. However, this approach ignores the importance differences of the pairwise interactions, resulting in the poor robustness. Nevertheless, one of the drawbacks to mean field theory is that it does not properly account for fluctuations when few interactions exist(Uzunov, 1993) (e.g., the average action may change drastically if there are only two adjacent agents). Wang et al. (2022) attempt to improve the representational ability of the merged agent by assign weight to each pairwise interaction by its attention score. However, the observations of other agents are needed as input, making this method not practical enough in the real world. In addition, the attention score is essentially a correlation in feature space, which seems

unconvincing. On the one hand, an agent pays more attention to another agent not simply because of the higher correlation. On the other hand, it may be inevitable that the proximal agents will be assigned high weight just because of the high similarity of their observation.

In this paper, we want to discuss a better way to represent the merged agent. We propose a algorithm named causal mean-field Q-learning (CMFQ) to address the shortcoming of MFQ in robustness via causal inference. Research in psychology reveals that humans have a sense of the logic of intervention and will employ it in a decision-making context(Sloman & Lagnado, 2015). This suggests that by allowing agents to intervene in the framework of mean-field reinforcement learning (MFRL), they could have the capacity to identify more essential interactions as humans do. Inspired by this insight, we assume that different pairwise interactions should be assigned different weights, and the weights could be obtained via intervening. We introduce a structural causal model (SCM) that represents the invariant causal structure of decision-making in MFRL. We intervene on the SCM such that the corresponding effect of specific pairwise interaction can be presented by comparing the difference before and after the intervention. Intuitively, the intervening enable agents to ask "what if the merged agent was replaced with an adjacent agent" as illustrated in Fig.1. In practice, the pairwise interactions could be embodied as actions taken between two agents, therefore the intervention also performs on the action in this case.

CMFQ is based on the assumption that the joint Q-function could be factorized into local pairwise Q-functions, which mitigates the dimension curse in the scalability problem. Moreover, CMFQ alleviates another challenge in the scalability problem, namely non-stationarity, by focusing on crucial pairwise interactions. Identifying crucial interactions is based on causal inference instead of attention mechanism. Surprisingly, the scalability performance of CMFQ is much better than the attention-based method(Wang et al., 2022). The reasons will be discussed in experiments section. As causal inference only needs local pairwise Q-functions, CMFQ is practical in real-world applications, which are usually partially observable. We evaluate CMFQ in the cooperative predator-prey game and mixed cooperative-competitive battle game. The results illustrate that the scalability of CMFQ significantly outperforms all the baselines. Furthermore, results show that agents controlled by CMFQ emerge with more advanced collective intelligence.

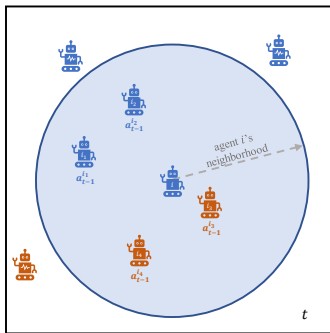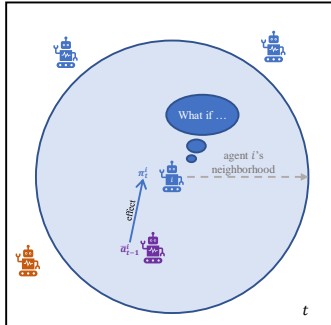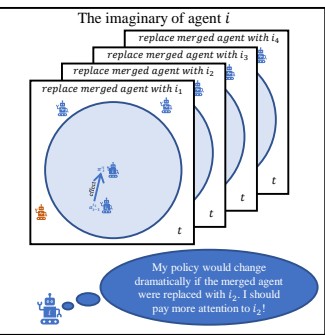

Figure 1: Blue agents and orange agents belong to different teams. The purple agent denote a merged agent that simply average all agents in agent $i$'s neighborhood. The diagram on the left shows a scenario in which the central agent $i$ interacts with many agents, $i_k$ denotes the $k^{th}$ agent in the observation of agent $i$. In the framework of MFRL, the scenario is transferred to the diagram in the middle, in which an merged agent is used to characterize all the agents in the central agent's observation. Our method further enables the central agent to learn to ask "what if". When it asks this question, it can imagine the scenario illustrated in the right diagram. The central agent can hypothetically replace the action of the merged agent in MFRL with the action of a neighborhood agent, and if this replacement will cause dramatic changes in policy, it means this neighborhood agent is potentially important. Thus central agent should pay more attention to the interaction with this neighborhood agent.

## 2 RELATED WORK

The scalability problem has been widely investigated in current literatures. Yang et al. (2018) propose the framework of MFRL that increases scalability by reducing the action-state space. Several works in a related area named mean-field game also proves that using a compact representation to

characterize population information helps solve scalability problem(Guo et al., 2019; Perrin et al., 2021).

Several works were proposed to improve MFQ. Wu et al. (2022) proposed a weighted mean-field assigning different weights to neighbor actions according to the correlations of the hand-craft agent attribute set, which is difficult to generalize to different environments. Wang et al. (2022) calculate the weights with attention score. The observations of other agents are needed to calculate the attention scores, making its practicality not satisfactory.

Our work is also closely related to recent development in causal inference. Researches indicate that once the SCM, which implicitly contains the causal relationships between variables, is constructed, we can obtain the causal effect by intervening. The causal inference has already been exploited for communication pruning(Ding et al., 2020), solving credit assignment problem(Foerster et al., 2018; Omidshafiei et al., 2019), demonstrating the potential of causal inference in reinforcement learning. (Pearl, 2019; 2001; Peters et al., 2017). Xia et al. (2021) and Zečević et al. (2021) further proved that SCM could be equally replaced with NCM under certain constraints, enabling us to ask "what if" by directly intervening on neural network.

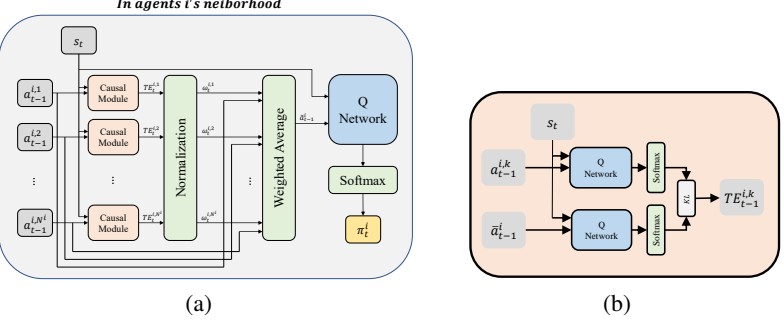

(a)  (b)

Figure 2: (a) is CMFQ's architecture. Each neighborhood agent is assigned a weight according to its causal effect to the policy of the central agent. (b) is the causal module. It calculate the $KL$ divergence between the two policies that the merged agent is represented by the average action and the $k^{th}$ neighborhood agent action respectively. A large $KL$ divergence means the $k^{th}$ neighborhood agent might be ignored in the merged agent represented by the average action, hence it should be assigned a higher weight to form a better merged agent.

## 3 PRELIMINARY

This section discusses the concepts of the stochastic game, mean-field reinforcement learning, and causal inference.

### 3.1 STOCHASTIC GAME

A $N$-player stochastic game could be formalized as $G = <S, \boldsymbol{A}, P, \boldsymbol{r}, N, \gamma>$, in which $N$ agents in the environment take action $\boldsymbol{a} \in \boldsymbol{A} = \times_{i=1}^{N} A^i$ to interact with other agents and the environment. Environment will transfer according to the transition probability $P(s' \mid s, \boldsymbol{a}) : S \times \boldsymbol{A} \times S \to [0, 1]$, then every agent obtains its reward $r^i(s, a^i) : S \times A^i \to \mathbb{R}$ and $\gamma \in [0, 1]$ is the discount factor. Agent makes decision according to its policy $\pi^i(s) : S \to \Omega(A^i)$, where $\Omega(A^i)$ is a probability distribution over agent $i$'s action space $A^i$.

The joint Q-function of agent $i$ is parameterized by $\theta_i$ and takes $s$ and $\boldsymbol{a}$. It is updated as

$$\mathcal{L}_i(\theta_i) = \mathbb{E}_{s,\boldsymbol{a},r,s'} \left[ \left( Q^i(s, \boldsymbol{a}; \theta_i) - y \right)^2 \right],$$
$$y = r + \gamma \max_{a'^i} Q^i\left( s', \boldsymbol{a}; \theta_i^- \right) \tag{1}$$

where $\theta_i^-$ is updated by with $\theta_i$ every $C$ steps and set fixed until the next $C$ steps finish.

where predict network parameters $\theta$ update at every epoch. Target network parameters $\theta^-$ denotes parameters to be updated with

### 3.2 MEAN FIELD REINFORCEMENT LEARNING

Mean field approximation turns a many-agent problem into a two-agent problem by mapping the joint action space to a single action space. The joint action Q function is firstly factorized considering only local pairwise interactions, then pairwise interactions are approximated using the mean-field theory

$$Q^i\left(s, a^1, a^2, \ldots, a^N\right) = \frac{1}{N^i} \sum_{k \in N(i)} Q^i\left(s, a^i, a^k\right) \approx Q^i\left(s, a^i, \bar{a}^i\right) \tag{2}$$

where $N^i = |N(i)|$. $N(i)$ is the set of agent $i$'s neighboring agents. Interactions between central agent $i$ and its neighbors are reduced to the interaction between the central agent and an abstract agent, which is presented by average behavior information of agents in the neighborhood of agent $i$. Finally, the policy of the central agent $i$ is determined by pairwise Q-function

$$\pi_t^i\left(a_t^i \mid s, \bar{a}_t^i\right) = \frac{\exp\left(\beta Q_t^i\left(s_t, a_t^i, \bar{a}_{t-1}^i\right)\right)}{\sum_{a'^i \in \mathcal{A}^i} \exp\left(\beta Q_t^i\left(s_t, a'^i, \bar{a}_{t-1}^i\right)\right)} \tag{3}$$

It is proven that $\pi_t^i$ will converge eventually(Yang et al., 2018).

### 3.3 CAUSAL INFERENCE

The data-driven statistical learning method lacks the identification of causality which is quite a vital part of composing human acknowledge. The SCM established with human knowledge is needed to represent the causality among all the variables we consider. An SCM is a 4-tuple $\mathcal{M} = < \mathbf{U}, \mathbf{V}, \mathbf{F}, P(\mathbf{U}) >$. $\mathbf{U} = \{U_1, U_2, \cdots, U_m\}$ is the set of exogenous variables which are determined by factors outside the model. $\mathbf{V} = \{V_1, V_2, \cdots, V_n\}$ is the set of endogenous variables that are determined by other variables. $\mathbf{F}$ is a set of functions $\{f_{V_1}, f_{V_2}, \cdots, f_{V_n}\}$ such that $f_{V_j}$ maps $\mathbf{Pa}_{V_j} \cup \mathbf{U}_{V_j}$ to $V_j$. where $\mathbf{U}_{V_j} \subseteq \mathbf{U}$ is all the exogenous variables directly point to $V_j$ and $\mathbf{Pa}_{V_j} \subseteq \mathbf{V} \backslash V_j$ is all the endogenous variables directly point to $V_j$. That is, $V_j = f_{V_j}(\mathbf{Pa}_{V_j}, \mathbf{U}_{V_j})$ for $j = 0, 1, \cdots, n$. $P(\mathbf{U})$ is the probability distribution function over the domain of $\mathbf{U}$. The causal mechanism in SCM $\mathcal{M}$ induced an acyclic graph $\mathcal{G}$, which uses a direct arrow to present a direct effect between variables as shown in Fig.3. Intervention is performed through an operator called $do(x)$, which directly deletes $f_X$ and replaces it with a constant $X = x$, while the rest of the model keeps unchanged. The equation defines the post-intervention distribution

$$P_{\mathcal{M}}(y|do(x)) \triangleq P_{\mathcal{M}_x}(y) \tag{4}$$

where $\mathcal{M}_x$ is the SCM after performing $do(x)$. Once we obtain the post-intervention distribution, one may measure the causal effect by comparing it with the pre-intervention distribution. A common measure is the average causal effect.

$$E[Y|do(x_0')] - E[Y|do(x_0)] \tag{5}$$

where $x_0'$ and $x_0$ are two different interventions. The causal effect may also be measured by the experimental Risk Ratio(Pearl, 2010)

$$\frac{E[Y|do(x_0')]}{E[Y|do(x_0)]} \tag{6}$$

## 4 METHOD

### 4.1 CONSTRUCTION OF SCM

The first step to obtain the causal effects behind the interactions among agents is constructing an SCM, which reveals the causal relations among all the variables. In the setting of MFRL, mean action $\bar{a}_{t-1}^i$ and state $s_t$ determine the policy $\pi_t^i(\cdot \mid s_t, \bar{a}_{t-1}^i)$ of agent $i$. As the key causal effect we concern is how important an interaction is for decision making, that is, how the interaction contains in $\bar{a}_{t-1}^i$ affects $\pi_t^i$, we construct the SCM center on $\pi_t^j$ as illustrated in Fig.3(b). The importance of the interaction with adjacent agent $k$ on the central agent $i$ could be estimated by replacing $\bar{a}_{t-1}^i$ with $a_{t-1}^{i,k}$ and quantified by the change of $\pi_t^i$. Formally, the causal effect of acting $a^k$ on $\pi_t^i$ is

$$TE_t^{i,k} = DIST(\pi_t^i(\cdot \mid s_t, a_t^i, \bar{a}_{t-1}^i), \pi_t^i(\cdot \mid s_t, a_t^i, do(\bar{a}_{t-1}^i = a_{t-1}^{i,k}))) \tag{7}$$

where $a_{t-1}^{i,k}$ is the action of the $k^{th}$ agent in the neighborhood of agent $i$. The causal effects in Eq.(5) and Eq.(6) are quantified using the difference in statistics before and after the intervention. As the policy are known, we can utilize the difference in policy to quantify causal effects. $DIST$ measures the difference between pre-intervention policy and post-intervention policy. We use $KL$ divergence as the $DIST$ function in practice. As $\pi_t^i$ is parameterized using neural network, the $do(\cdot)$ calculation is performed by directly changing the input of $\pi_t^i$. It is worth noting that not all neural networks are capable of causal inference(Xia et al., 2021). As a neural network learned by interacting with the environment, $\pi_t^i$ lies on the second layer of *Pearl Causal Hierarchy* (Bareinboim et al., 2022), and naturally contains both the causality between agent-wise interaction and the causality between agent-environment interaction. It is sufficient for estimating the causal effect of certain interaction.

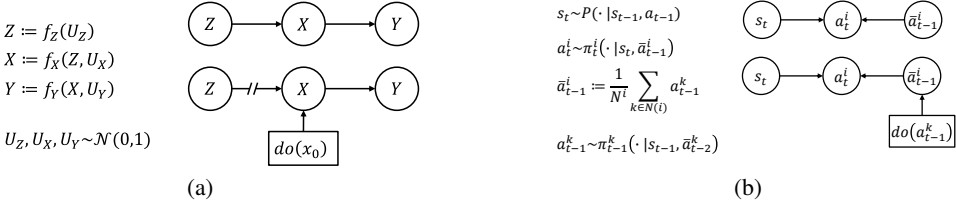

(a)                                                                 (b)

Figure 3: (a) is a canonical SCM, when $do(x_0)$ operation is performed on $X$, all causes of $X$ will be broken and keep all variable constant but only change $X$ to $x_0$. (b) is the SCM of MFRL, the $do(\cdot)$ operation on $\bar{a}_{t-1}^i$ follows the same procedure.

## 4.2 Improving MFQ with Causal Effect

In MFRL, we assume that different pairwise Q-functions should be assigned different weights depending on their potential influences on the policy of central agent. Hence, the factorization of Eq.(2) should be revised to

$$Q^i\left(s, a_t^1, a_t^2, \ldots, a_t^N\right) = \sum_{k \in N(i)} w^{i,k} Q^i\left(s, a^i, a^{i,k}\right) \tag{8}$$

where $N(i)$ is the set of agent $i$'s adjacent agents. Then $Q^j(s, a^1, a^2, \cdots, a^N)$ is approximated using mean-field theory and considering the causality-aware weights

$$Q^i(s, a^1, a^2, \cdots, a^N) = \sum_{k \in N(i)} w^{i,k} Q^i\left(s, a^i, a^{i,k}\right)$$

$$= \sum_{k \in N(i)} w^{i,k} \left[ Q^i\left(s, a^i, \breve{a}^i\right) + \nabla_{\breve{a}^i} Q^i\left(s, a^i, \breve{a}^i\right) \cdot \delta a^{i,k} + \frac{1}{2} \delta a^{i,k} \cdot \nabla_{\tilde{a}^{i,k}}^2 Q^i\left(s, a^i, \tilde{a}^{i,k}\right) \cdot \delta a^{i,k} \right]$$

$$= Q^i\left(s, a^i, \breve{a}^i\right) + \nabla_{\breve{a}^i} Q^i\left(s, a^i, \breve{a}^i\right) \cdot \left[ \sum_{k \in N(i)} w^{i,k} \delta a^{i,k} \right] + \sum_{k \in N(i)} w^{i,k} \left[ \frac{1}{2} \delta a^{i,k} \cdot \nabla_{\tilde{a}^{i,k}}^2 Q^i\left(s, a^i, \tilde{a}^{i,k}\right) \cdot \delta a^{i,k} \right]$$

$$= Q^i\left(s, a^i, \breve{a}^i\right) + \sum_{k \in N(i)} w^{i,k} R_{s,a^i}^i\left(a^{i,k}\right) \approx Q^i\left(s, a^i, \breve{a}^i\right) \tag{9}$$

where $\delta a^{i,k} = a^{i,k} - \breve{a}^i$ and $\breve{a}^i = \sum_{k \in N(i)} w^{i,k} a^{i,k}$, hence $\sum_k w^{i,k} \delta a^{i,k} = 0$. In the second-order term, $\tilde{a}^{i,k} = \breve{a}^i + \epsilon^{i,k} \delta a^{i,k}$, $\epsilon^{i,k} \in (0,1)$. $R_{s,a^i}^i\left(a^{i,k}\right)$ denotes the first-order Taylor expansion's Lagrange remainder which is bounded by $[-L, L]$ in the condition that the $Q^i\left(s, a^i, a^{i,k}\right)$ function is $L$-smoothed. To be self-contained, we put the derivation in Appendix B. The remainder is a value fluctuating around zero. As Yang et al. (2018) discussed in their work, under the assumption that fluctuations caused by adjacent agents tend to cancel each other, the remainder could be neglected. Once causal effects of pairwise interactions are known, the next question is how to to improve the representational capacity of the merged agent. Both linear methods, e.g., weighted sum, or nonlinear methods, e.g., encoding with a neural network, might be useful. However, to ensure the merged agent's reasonability, we prefer a representation in the linear space formed by adjacent agents' action vectors. An intuitive method that can induce reasonable output is a weighted sum. In

practice, we find that weighted sum using respective causal effects as weight is enough to effectively improve the representational capacity of average action

$$\pi_t^i \left( a_t^i \mid s_t, \check{a}_{t-1}^i \right) = \frac{\exp \left( \beta Q_t^i \left( s_t, a_t^i, \check{a}_{t-1}^i \right) \right)}{\sum_{a'^i \in \mathcal{A}^i} \exp \left( \beta Q_t^i \left( s_t, a'^i, \check{a}_{t-1}^i \right) \right)}, \quad \check{a}_{t-1}^i = \sum_{k \in N(i)} w_t^{i,k} a_{t-1}^{i,k} \qquad (10)$$

$$w_t^{i,k} = \frac{TE_t^{i,k} + \epsilon}{\sum_{k \in N(i)} \left( TE_t^{i,k} + \epsilon \right)} \qquad (11)$$

where subscripts are used to denote time steps. $TE_t^{i,k}$ is calculated according to Eq.(7). Each $a_{t-1}^{i,k}$ is encoded in one hot vector. Hence the weighted sum returns a reasonable representation in the linear space formed by the actions of neighborhoods. Moreover, the representation is close to essential actions, emphasizing high-potential impact interactions. A term $\epsilon$ was introduced to smooth the weight distribution across all adjacent agents, avoiding additional non-stationarity during training. Besides, the naive mean-field approximation could be achieved when $\epsilon \to \infty$.

The Q-function $Q^i$ update using the following loss function similar with Eq.(1)

$$\mathcal{L}_i(\theta_i) = \mathbb{E}_{s,\boldsymbol{a},r,s'} \left[ \left( Q^i \left( s, a^i, \check{a}^i; \theta_i \right) - \left( r + \gamma \max_{a'^i} Q^i \left( s', a'^i, \check{a}^i; \theta_i^- \right) \right) \right)^2 \right] \qquad (12)$$

## 5 EXPERIMENTS

We evaluate CMFQ in two tasks: a mixed cooperative-competitive battle game and a cooperative predator-prey game. In the battle task, we compare CMFQ with IQL(Tampuu et al., 2017), MFQ(Yang et al., 2018), and Attention-MFQ(Wang et al., 2022) to investigate the effectiveness and scaling capacity of CMFQ. We further verify the effectiveness of CMFQ in another task. In the predator-prey task, we compare CMFQ with MFQ and Attention-MFQ. Our experiment environment is MAgent(Zheng et al., 2018).

### 5.1 MIXED COOPERATIVE-COMPETITIVE GAME

**Task Setting.** In this task, agents are separated into two groups, each containing $N$ agents. Every agent tries to survive and annihilate the other group. Ultimately the team with more agents surviving wins. Each agent obtains partial observation of the environment and knows the last actions other agents took. Agents will be punished when moving and attacking to lead agents to act efficiently. Agents are punished when dead and only rewarded when killing the enemy. The reward setting requires the agent to cooperate efficiently with teammates to annihilate enemies. In the experiments, we train CMFQ, IQL, MFQ, and Attention-MFQ in the setting of $N = 64$, then we change $N$ from 64 to 400 to investigate the scalability of CMFQ. The concrete reward values are set as follow: $r_{attack} = -0.1, r_{move} = -0.005, r_{dead} = -0.1, r_{kill} = 5$. We train every algorithm in *self-play* paradigm.

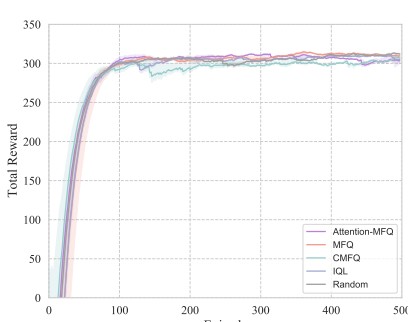

Figure 4: Total reward during training.

**Quantitative Results and Analysis.** As illustrated in Fig.5(a), we compare CMFQ with Attention-MFQ, MFQ, and IQL. We do not choose Wu et al. (2022) as a baseline because it is a correlation-based algorithm identical to Attention-MFQ. We assume that the attention-based method is a more challenging baseline. Moreover, in addition to these algorithms, we also set ablation algorithms named Random to verify that the performance improvement of CMFQ is not caused by randomization. Random follows the same pipeline as CMFQ but returns a random causal effect for each interaction. Fig.4 shows the learning curve of all algorithms. We can see that the total rewards of all algorithms converge to a stable value, empirically demonstrating the training scalability of our

algorithm.

To compare the performance of each algorithm, we put trained algorithms in the test environment that $N = 64$, and let them battle against each other. Fig.5(a) shows that MFQ performs better than IQL but worse than Attention-MFQ, indicating that the mean-field approximation mitigates the scalability problem in this task. However, the simply averaging as MFQ is not a good representation of the population behavioral information. In order to improve its representational ability for large-scale scenarios, it is necessary to assign different weights to different agents. Moreover, CMFQ outperforms Attention-MFQ during the test, verifying the correctness of our hypothesis that correlation-based weighting is insufficient to catch the essential interactions properly, while the intervention fills this gap by giving agents the ability to ask "what if".

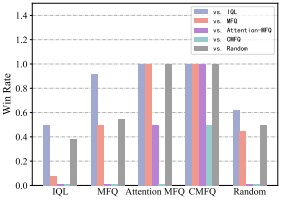
(a) Performance comparisons.

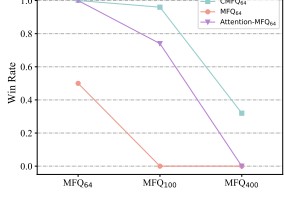
(b) Test Scalability curve.

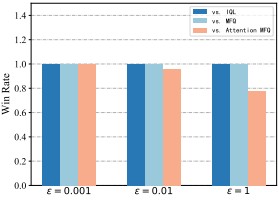
(c) Ablation experiments of $\epsilon$.

Figure 5: Win rate during execution. (a) shows the results that algorithms battle against each other. the horizontal axis is divided into five groups by algorithms, and within each group there are five bars representing the win rate of the algorithm on the horizontal axis. (b) shows win rates of algorithms in the label against MFQ algorithms which are on the horizontal axis. (c) shows the win rate of CMFQ with different $\epsilon$ against other algorithms.

We further investigate the test scalability of CMFQ, MFQ, and Attention-MFQ. Firstly, we train these three algorithms in 64 vs. 64 scenario with self-play, denoted as $\text{CMFQ}_{64}$, $\text{MFQ}_{64}$, $\text{Attention-MFQ}_{64}$ respectively, and further train the MFQ algorithm in 100 vs. 100 and 400 vs. 400 scenarios, denoted as $\text{MFQ}_{100}$ and $\text{MFQ}_{400}$. Then, allow $\text{CMFQ}_{64}$, $\text{MFQ}_{64}$, and $\text{Attention-MFQ}_{64}$ to battle against $\text{MFQ}_{64}$, $\text{MFQ}_{100}$ and $\text{MFQ}_{400}$ in environments 64 vs. 64, 100 vs. 100, 400 vs. 400 respectively, that is, letting CMFQ, MFQ, and Attention-MFQ control more agents than they were trained, to reveal the test scalability of the algorithms. As shown in Fig.5(b), the test scalability of MFQ is the worst, which means that we need to retrain MFQ when the number of agents increases and the test scalability of Attention-MFQ is slightly better. The test scalability of CMFQ is significantly better than both of them. Furthermore, CMFQ achieves win rates of nearly 100% against $\text{MFQ}_{100}$ and 32% against $\text{MFQ}_{400}$.

**Ablations.** We set two ablation experiments. The first one to ablate the effectiveness of causal effects in CMFQ. As illustrated in Fig.5(a), the performance of *Random* is inferior to MFQ, verifying the validity of causal effect in CMFQ. The other one is ablation for $\epsilon$. As we analyze in 4.2, $\epsilon$ is an adjustable parameter in the interval $[0, +\infty]$. As $\epsilon$ increases, the effect of each interaction becomes smoother and eventually CMFQ equal to MFQ when $\epsilon \to +\infty$. From the Fig.5(c), we can see that as we adjust $\epsilon$ from 0.001 to 1, the learning curve of CMFQ always converges, and in the test environment, win rates of CMFQ always outperform other baselines. When $\epsilon$ is relatively large, the win rate is close to that of MFQ.

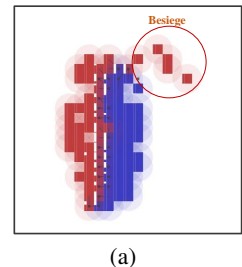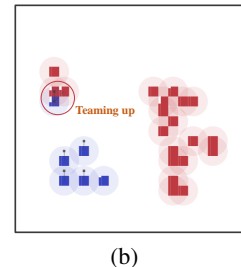

Figure 6: Visualization of CMFQ vs MFQ in 64 vs 64 environment. Red squares denote CMFQ, and blue squares denote MFQ, the vertical bar on the left side of the square indicates its health point, and the surrounding circular area indicates its attack range. When agent attacks, an arrow will be extended to point at the attack target.

**Visualization Analysis.** As illustrated in Fig.6(a), CMFQ learns the tactic of besieging, while MFQ tends to confront frontally. The results in Fig.6(b) indicate the tricky issue in mixed cooperative-competitive game: agents need to cooperate with their teammates to kill enemies, whereas only the agent who hits a fatal attack gets the biggest reward $r_{kill}$, driving agents hesitating to attack first. When there are few agents, the policies of MFQ and CMFQ tend to be conservative. However, CMFQ presents more advanced tactics: agents learn the trick of teaming up in the mixed cooperative-competitive game. When an agent chooses to attack, the adjacent teammates will arrive to help, achieving the maximum reward with the smallest cost of health. Moreover, Fig.6(b) also shows that attacks of CMFQ are more focused than baselines. CMFQ can discriminate key interactions and have a more accurate timing of attacks, while MFQ lacks this discriminatory ability and thus keeps attacking.

## 5.2 COOPERATIVE GAME

**Task Setting.** In this task, agents are divided into predator and prey. Prey move 1.5 times faster than predators, and their task is to avoid predators as much as possible. Predators are four times larger than prey and can attack but not yield any damage. Predators only get rewarded when they are close to prey. Therefore, to gain the reward, they must cooperate with other predators and try to surround prey with their size advantage. In our experiments, to test the scalability of the CMFQ, we first train MFQ, CMFQ, and Attention-MFQ employing the *self-play* paradigm in a scenario involving 20 predators and 40 prey, and then test them in

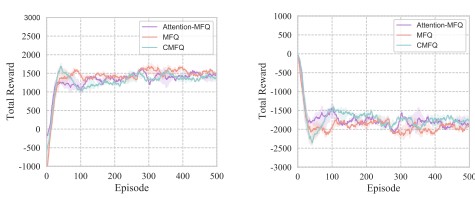

(a) Total reward of Preda- (b) Total reward of Prey.
tor.

Figure 7: Total reward during training.

environments involving (20 predators, 40 prey), (80 predators, 160 prey), (180 predators, 360 prey) respectively. The reward are set as follow: $r_{attack} = -0.2, r_{surround} = 1, r_{be\_surrounded} = -1$.

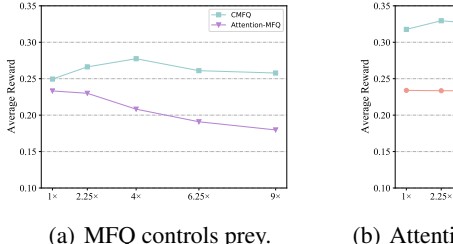 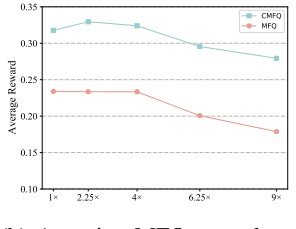 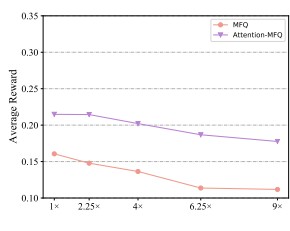

(a) MFQ controls prey.     (b) Attention-MFQ controls prey.     (c) CMFQ controls prey.

Figure 8: Total reward of predators during execution changes when the number of agents increases. $1\times$ denotes $N_{predator}$=20, $N_{prey}$=40, $4\times$ demotes $N_{predator}$=80, $N_{prey}$=160 and so on. All algorithms are trained in the $1\times$ environment.

**Quantitative Results and Analysis.** We compare CMFQ with MFQ and Attention-MFQ. First, we investigate their training scalability in (20 predators, 40 prey), as shown in Fig.7(a) and Fig.7(b), all of them converge to a stable reward total reward, verifying their training scalability. Then, we enlarge the number of agents during execution to investigate their test scalability. To demonstrate the scalability gap of different algorithms, we allow the algorithms to execute in an adversarial form, which means that one algorithm controls the predator and another controls the prey. For the environment, we change the number of agents to 1x, 4x, and 9x of the number in the training environment.

Because the reward $r_{be\_surrounded}$ of prey and the reward $r_{surround}$ of predator are zero-sum and cooperation exists mainly among predators, we use the total reward of predators to indicate each algorithm's performance. The results are shown in Fig.10. Total rewards in specific environment indicate the train scalability, since a higher total reward means agents learn better policy during

training. Trends of lines are related to test scalability, and a more flat line indicates the better test scalability of the algorithm. We can see that the total reward of Attention-MFQ is higher than that of MFQ, and the trend is similar to that of MFQ. In comparison, the total reward of CMFQ is higher than that of both MFQ and Attention- MFQ, and the trend is ever more flat, indicating that CMFQ has better scalability.

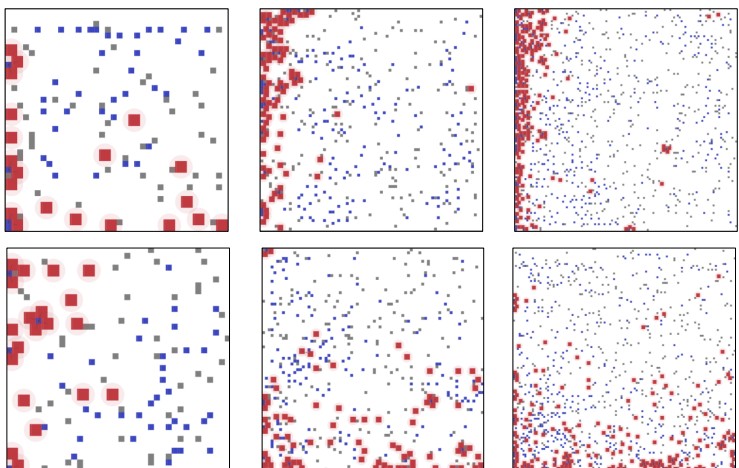

Figure 9: Visualization of cooperative predator prey game. The first row is results of CMFQ, the second row is results of Attention-MFQ. $N_{predator}$=20,$N_{prey}$=40 for the left column, $N_{predator}$=40,$N_{prey}$=80 for the middle column, $N_{predator}$=180,$N_{prey}$=360 for the last column. Red squares are predators while blue squares are prey, the grey squares are obstacles. All images are obtained 400 steps after the game begin.

**Visualization Analysis.** The results that the trained CMFQ and Attention-MFQ controls predators are shown in Fig.9. In the the environment that $N_{predator}$=20, $N_{prey}$=40, both CMFQ and Attention-MFQ perform similarly. Predators learn two strategies: four predators cooperating to surround the prey in an open area; two or three predators surrounding the prey with the help of obstacles. In the environment that $N_{predator}$=40, $N_{prey}$=80, when the number of agents increases, predators controlled by Attention-MFQ are more dispersed than predators controlled by CMFQ. Besides, MFQ has more predators idle than CMFQ. Predators controlled by CMFQ gather on map edges, because it is more efficient to surround prey with the help of map edges. In addition, predators controlled by CMFQ learn an advanced strategy to drive prey to map edges then take advantage of the terrain to surround them. In the environment that $N_{predator}$=180, $N_{prey}$=360, the advanced strategy is also presented. Moreover, predators controlled by CMFQ master the skill to utilize the bodies of still teammates who have captured prey as obstacles. Thus, predators controlled by CMFQ present a high degree of aggregation and environmental adaptability.

## 6 CONCLUSIONS

This paper aims at scalability problem in large-scale MAS. Firstly, We inherit the framework of MFRL which significantly reduce the dimensionality of joint state-action space. To further handle the intractable non-stationarity when the number of agent is large, we propose a SCM to model the decision-making process, and enable agents to identify the more crucial interactions via intervening on the SCM. Finally a causality-aware representation of population behavioral information could be obtained by the weighted sum of the action of each agent according to its causal effect. Experiments in two tasks reveal the excellent scalability of CMFQ.

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

## A  IMPLEMENTATION DETAILS

The pseudocode of CMFQ is listed below.

---
**Algorithm 1** Causal Mean Field Q-learning
---
**Input:**  Initialize state $s_0$; $Q_{\theta_i}, Q_{\theta_i^-}, \check{a}_0^i$ for all agent $i \in \{1, 2, \cdots, N\}$; trajectory length $M$;
  **while** in the training loop **do**
    **for** $t = 0, 1, \cdots, M$ **do**
      **for** $i = 1, 2, \cdots, N$ **do**
        Calculate policy $\pi_t^i(\cdot \mid s_t, \bar{a}_{t-1}^i)$ with average merged agent;
        Calculate causal effect for every neighborhood agent by Eq.(7);
        Obtain a new merged agent $\check{a}_{t-1}^i$ and a new policy $\pi_t^i(\cdot \mid s_t, \check{a}_{t-1}^i)$ by Eq.(10);
      **end for**
      Sample joint action $\mathbf{a} = [a^1, a^2, \cdots, a^N]$ from $[\pi_t^1, \pi_t^2, \cdots, \pi_t^N]$
      obtain the next state $s_{t+1}$ and the reward $\mathbf{r} = [r^1, r^2, \cdots, r^N]$ and merged agent $\check{\mathbf{a}} = [\check{a}_{t-1}^1, \check{a}_{t-1}^2, \cdots, \check{a}_{t-1}^N]$;
      Store transition $< s_t, \mathbf{a}, \mathbf{r}, s_{t+1}, \check{\mathbf{a}} >$ in replay buffer;
    **end for**
    **for** $i = 1, 2, \cdots, N$ **do**
      Sample a minibatch transition from replay buffer;
      Calculate $\mathcal{L}_i$ and update $\theta_i$ by Eq.(12);
      Updata target network by $\theta_i^- = \theta_i$ after every $C$ updates of $\theta_i$;
    **end for**
  **end while**
---

## B  DERIVATION FOR THE BOUND OF THE LAGRANGE REMAINDER

As $s, a^i$ in $Q^i(s, a^i, a^{i,k})$ are fixed parameter in the derivation of Eq.(9), for simplicity, the pairwise Q-function $Q^i(s, a^i, a^{i,k})$ can be rewrite as $Q(a^k)$ in the following. We assume that $a^k$ is a one-hot encoding for n actions, to make $Q(a^k)$ more general, we replace the discrete $a^k$ ($a^k \in \mathbb{R}^N$) by a continuous $x$ ($x \in \mathbb{R}^N$) which don't violate the domain of the parameterized Q-function. Given the $Q(x)$ is $L$-smooth, then for any two points $x, y \in dom(Q) \subseteq \mathbb{R}^N$, there exists a Lipschitz constant $L \in [0, +\infty)$ that

$$\|\nabla Q(x) - \nabla Q(y)\|_2 < L\|x - y\|_2 \tag{13}$$

By the first order Taylor expansion with Lagrange remainder, we have

$$\nabla Q(y) = \nabla Q(x) + \nabla^2 Q(x) \cdot u + R(u) \tag{14}$$

where $u = y - x, \lim_{u \to 0} \frac{R(u)}{\|u\|_2} = 0$. Assume $x \neq y$, then we can reform the first order Taylor expansion

$$\begin{aligned}
\frac{\|\nabla^2 Q(x) \cdot u\|_2}{\|u\|_2} &= \frac{\|\nabla Q(y) - \nabla Q(x) - R(u)\|_2}{\|u\|_2} \\
&\leq \frac{\|\nabla Q(y) - \nabla Q(x)\|_2}{\|u\|_2} + \frac{\|R(u)\|_2}{\|u\|_2} \\
&\leq L + \frac{\|R(u)\|_2}{\|u\|_2}, \quad \forall x, y \in dom(Q), x \neq y
\end{aligned} \tag{15}$$

$u$ could be the eigenvalue of $\nabla^2 Q(x)$, then Eq.(15) can be convert to

$$\frac{\|\nabla^2 Q(x) \cdot u\|_2}{\|u\|_2} = \frac{\|\lambda u\|_2}{\|u\|_2} = \mid \lambda \mid \leq L + \frac{\|R(u)\|_2}{\|u\|_2} \tag{16}$$

Obviously, we can obtain the bound of $\lambda$, $\lambda \in [-L, L]$. $\nabla^2 Q(x)$ is a real symmetric matrix, so there exist an orthogonal matrix U to diagonalize $\nabla^2 Q(x)$ such that $U^T [\nabla^2 Q(x)] U = \Lambda \triangleq diag[\lambda_1, \lambda_2, \ldots, \lambda_N]$. Then the bound of $R_{s,a^i}^i(a^{i,k})$ can be derived as follow

$$R_{s,a^i}^i(a^{i,k}) = \frac{1}{2}\delta a^{i,k} \cdot \nabla^2 Q(a^k) \cdot \delta a^{i,k} = \frac{1}{2}\left[U \cdot \delta a^{i,k}\right]^T \Lambda \left[U \cdot \delta a^{i,k}\right] = \frac{1}{2}\sum_{n=1}^{N} \lambda_n \left[U \cdot \delta a^{i,k}\right]_n^2 \tag{17}$$

$$-L\|U \cdot \delta a^{i,k}\|_2 \le \sum_{n=1}^{N} \lambda_n \left[U \cdot \delta a^{i,k}\right]_n^2 \le L\|U \cdot \delta a^{i,k}\|_2 \tag{18}$$

where $\left[U \cdot \delta a^{i,k}\right]_n$ refers to the $n^{th}$ element of vector $U \cdot \delta a^{i,k}$.

$$\begin{aligned}
\|U \cdot \delta a^{i,k}\|_2 = \|\delta a^{i,k}\|_2 &= (a^{i,k} - \check{a}^i)^T(a^{i,k} - \check{a}^i) \\
&= a^{i,k^T}a^{i,k} + \check{a}^{i T}\check{a}^i - \check{a}^{i T}a^{i,k} - \check{a}^i a^{i,k^T} = 2(1 - \check{a}_n^i) \le 2
\end{aligned} \tag{19}$$

where $a^{i,k}$ is a one-hot encoding action, $\check{a}_n^i$ denotes the $n^{th}$ element in $\check{a}^i$. Finally, according to Eq.(17) Eq.(18) Eq.(19), the bound of $R_{s,a^i}^i(a^{i,k})$ is $[-L, L]$.

## C  VISUALIZATION FOR THE WEIGHTS OF CMFQ AND ATTENTION-MFQ

To further analyze the reasons why CMFQ is more effective than Attention-MFQ empirically, we randomly select an agent in the mixed cooperative-competitive game task and visualize its weight. Some interesting observations can be made from Fig.10(a). First of all, it makes sense that the agents on the front line will be given high weights because they are battling. Secondly, the weights of agents at the edge of the front line are relatively small, possibly because these agents can cooperate with nearby teammates to attack an enemy due to their position advantages, so they are in a relatively dominant state. In addition, agents at the very edge of the front line are given higher weights, even if they are out of combat. This is because they are in a position to flank their opponents and work with their teammates to surround the opponents. In Fig.10(b), we observe a result consistent with the analysis in our paper. That is, the attention-based method uses the attributes of other agents to calculate the attention scores, and observation is an important part of the attributes, so it tends to give high weight to the agents nearby because their observations are similar.

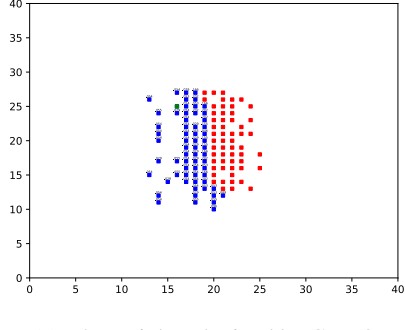 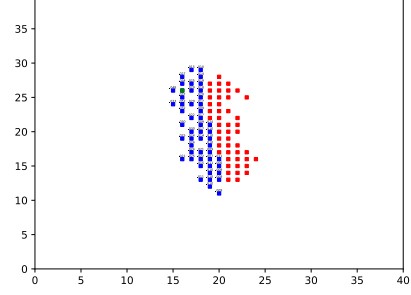

(a) The weights obtained by CMFQ.     (b) The weights obtained by Attention-MFQ.

Figure 10: The two figures visualize the mixed cooperative-competitive task, where each agent in the blue team in (a) is controlled by CMFQ and each agent in the blue team in (b) is controlled by Attention-MFQ. Each agent in the red team is controlled by MFQ. We label the agents in the blue team whose weights are visualized in green. The number above the blue agent represents the normalized weight given by the green agent to the pairwise interaction between them. Due to space constraints, the integer bits of all weights are omitted

# D SUPPLEMENTAL EXPERIMENT ON MPE

To further investigate the applicability of CMFQ, we perform an experiment on another environment named multi-agent particle environment (MPE)(Mordatch & Abbeel, 2017). As the dimentionality of action-state space will change as the initial number of agent changes, making it difficult to verify scalability, but we believe that CMFQ's scalability performance has been adequately validated in previous experiments. For MPE, we tested the predator prey task in MPE when the number of agents was the same as that in the training environment, and compared it with 5.2 to see whether the same conclusions could be drawn in the two environments.

**Task Setting.** There are 20 predators, 40 preys, and 20 obstacles. Predator gets $r_{collide} = 10$ if it collide with prey. Prey gets $r_{be\_collided} = -10$ if it collided with predator. The speed of prey is 1.3 times of that of predator. In order to make preys learn to leverage obstacles instead of running to infinity, we manually draws an area. If preys go beyond this area, they will get penalty $r_{bound}$ which will be aggravate as the distance preys go beyond this area increase, until $r_{bound} = -10$. We trained MFQ, CMFQ and Attention-MFQ in the self-play paradigm. The training curve is shown in Fig.11.

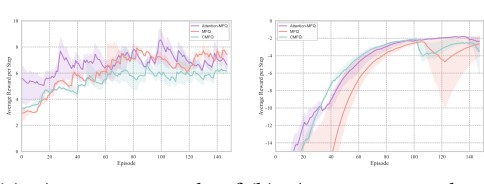

(a) Average reward of Predator.  (b) Average reward of Prey.

Figure 11: Average reward during training.

**Quantitative Results and Analysis**In the test phase, we controlled 20 Predators and 40 prey with different algorithms respectively, test 10 times and calculated the average reward of each algorithm, as shown in Table.1. First, the average reward of MFQ is lower than CMFQ and Attention-MFQ, regardless of whether it controls predators or preys. This indicates that the representational ability of average merged agent is insufficient. Secondly, when MFQ controls prey, the average predator reward of CMFQ is higher than Attention-MFQ, indicating that the weight obtained by CMFQ was more representational. Finally, in the comparison between CMFQ and Attention-MFQ, CMFQ outperforms Attention-MFQ in both predator reward and prey reward, further confirms the superiority of CMFQ. In the task that the number of agents in testing was the same as that in the training, We compare the performance of MFQ, CMFQ, and Attention-MFQ and come to the same conclusion consistent with 5.2, empirically certify the applicability of CMFQ.

| Predator | Predator reward | Prey | Prey reward |
|---|---|---|---|
| MFQ | 4.23 | CMFQ | -8.64 |
| CMFQ | 6.68 | MFQ | -13.05 |
| MFQ | 4.01 | Attention-MFQ | -8.47 |
| Attention-MFQ | 6.05 | MFQ | -12.46 |
| CMFQ | 3.15 | Attention-MFQ | -11.07 |
| Attention-MFQ | 3.02 | CMFQ | -4.23 |

Table 1: Results that let two different algorithm control predators and preys respectively. Predator reward is the average reward a predator obtain every step. Prey reward is the average reward a prey obtain every step.

