# OpenReview forum: "Causal Mean Field Multi-Agent Reinforcement Learning"
_ICLR.cc/2023/Conference — Submitted to ICLR 2023_

### Official Review · Reviewer_v11b · 2022-10-22

**Confidence:** 3
**Correctness:** 3
**Technical Novelty And Significance:** 4
**Empirical Novelty And Significance:** 4
**Recommendation:** 5

**Clarity, Quality, Novelty And Reproducibility:**

Clarity:  Good

Quality: Good

Reproducibility: Not sure

**Strength And Weaknesses:**

Strength:

1. I think the proposed method is quite novel and addresses an important issue
2. Empirical studies are good.



Weakness:
1. I think Figure 2need more explanation, I could understand the mechanism intuitively, but are there some more insightful explanation? Fig 2(b) is essentially comparing the KL divergence between two policies by using a concrete agent or a merged agent, the result in used in Fig 2(a) to generate the weight, why would this be a good mechanism?
2. Could you add more explanation to the L-smooth justification of the Q function with respect to action? I thought in your game, action is discrete?


**Summary Of The Paper:**

Multi-agent interactions result in non-stationary environment for reinforcement learning algorithms, which harms the performance of mean field method, which reduces high number agent's problem to pairwise interactions approximations.

To make the algorithm more robust, the authors propose learning counterfactual effects to generalized representation of merged agents.



Mean field method reduces the heterogeneous agents interactions to be centered to a  common agent interaction with its merged neighbors. Concretely, the value-action function involves the state of the agent in question and all its neighbors' interaction, which is decomposed to be the sum of pair wise value-action and approximated to be interaction with merged agent state-value function.


To quantify the importance of one specific neighbor to the current agent in question, the author proposes to quantify using the counterfactual policy distance by replacing the merged agent with that specific neighbor.
This importance weight is then used to average the contributions of the pairwise state-value function to the state value function with respect to all its neighbors.

The author further use Taylor expansion and assume state-value function is L-smooth thus the Lagrange remainder could be eliminated. The policy is conditioned on a weighted sum of previous step actions.


To evaluate the performance empirically,   IQL, MFQ(mean field) and attention MFQ is used as baselines.  Test environment has more agents than training environments.

**Summary Of The Review:**

I think the paper addresses an important issue in multi-agent reinforcement learning, where the number of agents are big and existing mean field method tend to ignore the different contributions of neighbors to the fate of the agent in question. The proposed method is well explained, though lack more in depth analysis.

---

> ### Author Response · Authors · 2022-11-17
> **Looking forward to your post-rebuttal feedback**
>
> Thank you for your insightful comments. We are glad that you find our work is “quite novel and addresses an important issue”, and our “empirical studies are good”. We address each of your concerns below.
> >I think Figure 2 need more explanation, I could understand the mechanism intuitively, but are there some more insightful explanation? Fig 2(b) is essentially comparing the KL divergence between two policies by using a concrete agent or a merged agent, the result in used in Fig 2(a) to generate the weight, why would this be a good mechanism?
>
> The mechanism in Fig.2 is actually mimicking the counterfactual inference that human will do during decision-making process [1]. We hope the following discussions could answer your question.
>
> **Why is KL a good quantification for causal effect?**
>
> Causal effect is essentially measuring how a variable change after intervening. When the distribution of variable is unknown, calculating expectations like Eq.(5) and Eq.(6) is a preferred quantification. In our case, however, we have already known the distribution of $a_t^i$ (which is exactly $π_t^i$), causal effect could be measured directly by KL divergence. Besides, Jangzing et. al. [2] have investigated the proper quantification for causal effect. They conclude that the KL divergence between the old and new distribution is a suitable measurement for causal effect. So KL divergence is not only a feasible quantificationfor causal effect, but also a good quantificationfor causal effect.
>
> **Why does larger KL divergence mean higher importance?**
>
> $π^i (\cdot|s,\bar{a}^i)$ can be understand as the policy when the central agent distribute attention evenly to neighborhood agents, and $π^i (\cdot|s,a^{i,k})$ could be seen as the policy when central agent pays all attention to the k-th neighborhood agent. A large KL divergence represents a significant difference between two policies, hence a preferred action in $π^i (\cdot│s,\bar{a}^i)$ could be a terrible action in $π^i (\cdot|s,a^{i,k})$. This is a risk signal that the k-th agent might be ignored in the average merged agent and a potentially bad action might be chosen. Hence, the central agent should pay more attention to the interaction with the k-th agent.
>
>
> In addition, we have visualized the weights of CMFQ and Attention-MFQ respectively in Appendix C for more in-depth analysis. The weights of CMFQ is more reasonable and explainable, while Attention-MFQ tends to assign high weights to nearby agents. Please check our latest manuscript for the visualization results.
> >Could you add more explanation to the $L$-smooth justification of the Q function with respect to action? I thought in your game, action is discrete?
>
> The discrete action doesn’t affect the $L$-smooth of Q-function, because action is just an independent variable of Q-function. As the Q-function parameterized by a NN is typically continuous and differentiable, $L$-smooth of Q-function is valid no matter action is continuous or discrete. We have reformed the derivation for the bound of the Lagrange remainder in Appendix B in the latest manuscript. Please check out our new manuscript to see the full proof.
>
> [1] Sloman, Steven A., and David Lagnado. "Causality in thought." Annual review of psychology 66 (2015): 223-247.
>
> [2] Janzing, Dominik, et al. "Quantifying causal influences." The Annals of Statistics 41.5 (2013): 2324-2358.
> \
> \
> _We sincerely appreciate your comments. Please let us know if you have further feedback._
>
> _Best, Authors_

---

> > ### Comment · Reviewer_v11b · 2022-12-09
> > **post-rebuttal feedback**
> >
> > Dear authors, thanks for your response,
> >
> > - To your response of using KL divergence to quantify causal effect: I agree, if a DAG is present, then one can use KL divergence to measure the distribution shift upon intervene.  But in your case, how do you know a hypothetical DAG is there in correspondence to the computational graph of your neural network?

---

> > > ### Author Response · Authors · 2022-12-09
> > > **Response to reviewer v11b**
> > >
> > > Dear reviewer v11b,
> > >
> > > Thank you for your question. The SCM shown in Fig.3(b) contains two parts, a set of structural assignments on the left are derived from the modeling of the sequential decision problem by *Stochastic Game* and *Mean Field Reinforcement Learning*, while the DAG on the right portrays the generation process of variable $a_t^i$ based on its structural assignment. Hence, **we consider the DAG to be sound rather than hypothetical.**
> > >
> > > Moreover, if the structural assignment of a variable is unknown, generally a neural network may be needed to train on observation data to fit it. Whereas in our case, the structural assignment of variable $a_t^i$ is already known. It is characterized by the Q network. That is, the policy output by the Q network is used both for decision making and to portray the distribution of variable $a_t^i$ in the DAG. Therefore, **the DAG is exactly corresponding to the computational graph of Q network.**
> > >
> > > In addition, you may also notice that according to the structural assignments on the left side of Fig.3(b), the complete DAG should contain all the variables in the entire sequential decision process (which is similar to the Fig.10.2 in Chapter.10 of the book *Elements of Causal Inference*, https://library.oapen.org/bitstream/id/056a11be-ce3a-44b9-8987-a6c68fce8d9b/11283.pdf). Since we only concern $a_t^i$ and only make counterfactual inferences around $a_t^i$, we think it is sufficient to model the structural assignment  as the DAG in Fig3(b).
> > >
> > >
> > > *Please let us know if you have any further question.*
> > >
> > > *Best, Authors*

---

### Official Review · Reviewer_bK4T · 2022-10-26

**Confidence:** 3
**Correctness:** 3
**Technical Novelty And Significance:** 3
**Empirical Novelty And Significance:** 2
**Recommendation:** 5

**Clarity, Quality, Novelty And Reproducibility:**

The paper is well written and builds on mean field q-learning. The code and settings are made available for reproducibility.



**Strength And Weaknesses:**

strengths:
-the paper address the limitation of ignoring pairwise interactions in mean field Q-learning using by using a weighted combination through causal inference
- factorizing joint-q function into local pairwise Q-functions is the basis to address scalability issues

weakness:
-more experimental scenarios need to be evaluated
-the figures are hard to read and need to be explained more clearly especially 5a
-how the proposed method improves scalability needs to be demonstrated in the experimental results when compared to the other approaches



**Summary Of The Paper:**

The paper proposes causal mean-field Q-learning (CMFQ) to address the scalability problems in non-stationary environments. Results show better scalability in mixed cooperative-competitive game and a cooperative game.

**Summary Of The Review:**

The paper builds on mean field q-learning hence the contribution may be limited. More experimental scenarios can be considered to showcase the superiority of the proposed approach.

---

> ### Author Response · Authors · 2022-11-17
> **Looking forward to your post-rebuttal feedback, part 1/2**
>
> Thank you for your constructive comments. We address each of your concerns with new experiments and clarifications below.
> >more experimental scenarios need to be evaluated.
>
> We completely understand your concern about the applicability of our method. The original experiments are performed on MAgent for two reasons. First, MAgent is suitable for evaluating algorithms in scenarios contain a large number of agents (e.g., in our experiments, a maximum of 800 agents are included). Besides, in MAgent, the dimensionality of joint action-state space remains unchanged as the number of agent increase, providing great convenience for evaluating test scalability. We believe that the experiments in this paper have demonstrated the superiority of CMFQ in improving scalability. But to further prove the applicability of our algorithm empirically, we add new experiments in multi-agent particle environment (MPE) which is another widely used environment for multi-agent reinforcement learning. We choose the predator prey task in MPE to facilitate the comparison with the results of predator prey task (with 20 predators and 40 preys) in MAgent. MFQ, Attention-MFQ, CMFQ are trained in self-play paradigm, then let the different algorithms control predator and prey separately and compare their rewards. Please check the Appendix D in our new manuscript for more analysis and training details. We put the results below.
> |    Predator   | Predator reward |  Prey | Prey reward |
> | :-------------: | :---------------: | :-------------: | :-----------: |
> | MFQ           | 4.23            | CMFQ          | -8.64       |
> | CMFQ          | 6.68            | MFQ           | -13.05      |
> | MFQ           | 4.01            | Attention-MFQ | -8.47       |
> | Attention-MFQ | 6.05            | MFQ           | -12.46      |
> | CMFQ          | 3.15            | Attention-MFQ | -11.07      |
> | Attention-MFQ | 3.02            | CMFQ          | -4.23       |
>
> We find that CMFQ outperforms both MFQ and Attention-MFQ, which is consistent with the conclusions in MAgent.
>
> >the figures are hard to read and need to be explained more clearly especially 5a
>
> Thank you for your advice. We have reviewed all the images in our paper and revised some annotations in the latest manuscript. To facilitate understanding, we explain Fig.5(a) here specifically. In Fig.5(a), we have 5 algorithms trained on 64 vs. 64 mixed cooperative-competitive game task, and we want to compare their performance against each other (including themselves) on 64 vs. 64 mixed cooperative-competitive task. The horizontal axis is divided into five groups by algorithm, and within each group there are five bars representing the win rate of the algorithm on the horizontal axis against the other algorithms.
>
> >how the proposed method improves scalability needs to be demonstrated in the experimental results when compared to the other approaches
>
> CMFQ significantly improves the test scalability by improving the robustness of MFQ. This is demonstrated in Fig.5(b) and Fig.8. Below, we explain how CMFQ improved scalability according to the results in Fig.5(b).\
> In the experiment results demonstrated in fig.5(b), we have $CMFQ_{64}$ trained in the 64vs.64 battle task by self-play paradigm waiting for evaluation. Meanwhile, we have $MFQ_{64},MFQ_{100},MFQ_{400}$ trained in 64 vs. 64, 100 vs. 100, 400 vs. 400 battle tasks respectively as baselines. Then, we let $CMFQ_{64}$ battle against $MFQ_{64},MFQ_{100},MFQ_{400}$ in 64 vs. 64, 100 vs. 100, 400 vs. 400  environments respectively. This may sound challenging, but as demonstrated in Fig.5(b), $CMFQ_{64}$ shows remarkable test scalability and even achieve a win rate of 32% against $MFQ_{400}$ in 400 vs. 400 battle task. \
> To further analyze how CMFQ outperforms Attention-MFQ in detail, in the latest manuscript, we add visualization analysis for the weights of CMFQ and Attention-MFQ in Appendix C. As shown in Fig.10, Attention-MFQ tends to assign higher weight to adjacent agents. In contrast, the weights of CMFQ seems more reasonable and explainable. More visualization for weights can be found in the supplementary material.

---

> ### Author Response · Authors · 2022-11-17
> **Looking forward to your post-rebuttal feedback, part 2/2**
>
> >The paper builds on mean field Q-learning hence the contribution may be limited.
>
> We understand your concern, but we have to emphasize that mean field Q-learning (MFQ) does not fully address the scalability problem although it indeed proposes an elegant and promising framework called mean field reinforcement learning to partly mitigate the scalability problem. If we decompose the scalability problem into 1) the dimensionality of joint state-action space increases exponentially as agent number increases, and 2) the non-stationarity increases as agent number increases. We could find that MFQ only solves the first part of scalability problem while the second part is largely unresolved. The lacking of robustness for non-stationarity makes MFQ have a poor test scalability, which can be proved by our experiment results that when the number of agents increases in the test environments, MFQ fails rapidly.\
> To solve the second part of scalability problem and make the algorithm robust to the non-stationarity, we believe that a mechanism to identify crucial interactions is necessary. By mimicking the counterfactual inference humans make before making decision [1], we make agents do counterfactual inference before taking action, and assign weights to neighborhood agents by the causal effect. This mechanism helps central agents to identify a few crucial neighborhood agents that are important to the decision-making while the number of agents increases. Empirical results show the robustness of CMFQ for non-stationarity is significantly enhanced.
>
> [1] Sloman, Steven A., and David Lagnado. "Causality in thought." Annual review of psychology 66 (2015): 223-247.
> \
> \
> _We sincerely appreciate your comments. Please let us know if you have further feedback._
>
> _Best, Authors_

---

### Official Review · Reviewer_3dJB · 2022-10-30

**Confidence:** 3
**Correctness:** 3
**Technical Novelty And Significance:** 3
**Empirical Novelty And Significance:** 3
**Recommendation:** 6

**Clarity, Quality, Novelty And Reproducibility:**

Clarity:
For writing, the logical flow of this paper is clear. However, there are also some minor problems.
1. The meaning of F and fVj(.) in section 3.3 is not explained. And the definition of F is not clear and is nested with fVj.
2. What do the Q^j and Q^I indicate in section 4.2 respectively?
3. Why does the larger TE(KL divergence ) result in the larger weights in equation 11? I think that a smaller KL divergence would correspond to a more important action thus with a larger weight.
4. I am confused about how fig 6(a) shows that “CMFQ learns the tactic of besieging, while MFQ tends to confront frontally”.
Quality: This work proposes a new causal mean-field Q-learning algorithm, improving the traditional mean-field Q-learning by assigning different weights when combining Q functions. Experiments results are sufficient to prove the effectiveness of the casual mean-field algorithm on test scalability and reward performance.
Novelty: I think this approach is original. It introduces the causal inference to measure the importance of Q functions, which is different from the previous method and empirically effective.
Reproducibility: The primary procedures to computing casual mean-filed Q-function are given by equations, it’s clear. It would be clearer if more details and pseudocode of the algorithm CMFO are added.


**Strength And Weaknesses:**

Strength

1. Adequate experiments show better performance on reward and scalability than other baselines and surprising visualization analysis shows the CMFQ has better ability to cooperate.

2. This work employs the causal model to infer the weight of pairwise Q functions replacing previous average-based mean-field and attention mean-field methods. I think this approach is original.

Weaknesses:
1. In the introduction section, the potential shortcoming of previous mean-field Q-learning (MFQ) is discussed, however, the discussion about attention-MFQ, e.g. the limitation of attention-MFQ seems missing.
2. equation (9) seems to need more explanation, and what the $Q^j$ and $Q^i$ indicate respectively?


**Summary Of The Paper:**

This paper addresses the scalability problem in MARL, scaling to environments containing more agents during testing, and proposes a causal mean-field Q-learning (CMFQ) algorithm. This paper solves the non-stationarity challenge by focusing on crucial pairwise interactions identified by Causal inference. And Mean Field Theory is beneficial to scalability since it turns a many-agent problem into a two-agent problem. They evaluate CMFQ in the cooperative predator-prey game and mixed cooperative-competitive battle game. The results illustrate that the scalability of CMFQ significantly outperforms all the baselines. Furthermore, results show that agents controlled by CMFQ emerge with more advanced collective intelligence than those controlled by other baselines.


**Summary Of The Review:**

This work proposes a new causal mean-field Q-learning algorithm, it is original to apply causal inference in mean-field Q-learning. This paper addresses the scalability problem,  experiments results are sufficient to prove that it can scale to environments containing more agents. The writing of this paper is easy to follow, and the structure is clear.

---

> ### Author Response · Authors · 2022-11-17
> **Looking forward to your post-rebuttal feedback**
>
> Thank you for your positive comments. We are glad to hear that you find our work is “original” and experiments are “adequate”. We address each of your concerns below.
> >In the introduction section, the potential shortcoming of previous mean-field Q-learning (MFQ) is discussed, however, the discussion about attention-MFQ, e.g. the limitation of attention-MFQ seems missing.
>
> In the former manuscript, we put the discussion about Attention-MFQ in the related work section. We agreed with you that it would be better to place the discussion in the introduction section. The discussion has been moved to introduction section in the latest manuscript. For clarification, we retell the discussion below.\
> Existing Attention-MFQ method use the attention score between attributes (e.g. observations, actions) of two agents as the weight, which makes the method not practical enough in the real world. In addition, the attention score is essentially a correlation in feature space. It may be inevitable that adjacent agents are assigned high weights just because their observations are similar to the observation of central agent, although they are trivial. Furthermore, in decision-making process, human intuitively don’t judge the importance of an agent by the correlation. Actually, human do counterfactual inference before making decision to predict possible situations. We assume the weight should also be obtained by counterfactual inference as what human do.
>
> > equation (9) seems to need more explanation, and what the $Q^j$ and $Q^i$ indicate respectively?
>
> We have added more explanation for equation (9) in the latest manuscript. Equation (9) essentially is the first-order Taylor expansion of the pairwise Q-function at $\\check{a}^i$ with Lagrange remainder. The proof for the bound of Lagrange remainder is given in the Appendix B.\
> $Q^j$ is a typo, it should be $Q^i$. Thank you for pointing out this typo. We have revised it and checked the paper carefully, there should be no typo in the new submitting.
>
> > The meaning of $\\mathbf{F}$ and $f_{V_j}\\ (\\cdot)$ in section 3.3 is not explained. And the definition of $\\textbf{F}$ is not clear and is nested with $f_{Vj}$ .
>
> We have polished the according description in the latest manuscript. The value of each endogenous variable $V_j$ is determined by its structural equation, $f_{V_j}\ (Pa_{V_j}\ ,U_{V_j}\ )$, where $Pa_{V_j}\ $  is all the endogenous variables directly point to $V_j\ $, and $U_{V_j}\ $ is all the exogenous variables directly point to $V_j\ $. $\\mathbf{F}$ is the set of all structural functions $\\{f_{V_1}\ ,f_{V_2}\ ,...,f_{V_n}\ \\}$ in an SCM.
>
> >Why does the larger TE(KL divergence ) result in the larger weights in equation 11? I think that a smaller KL divergence would correspond to a more important action thus with a larger weight.
>
> $\pi^i(\cdot|s,\bar{a}^i)$ can be understand as the policy when the central agent distribute attention evenly to neighborhood agents, and $\pi^i(\cdot|s,a^k)$ could be seen as the policy when central agent pays all attention to the k-th neighborhood agent. A higher TE between the two policies means that a preferred action in $\pi^i(\cdot|s,\bar{a}^i)$ could be a bad choice in $\pi^i(\cdot|s,a^k)$. This is a risk signal that the k-th neighborhood agent may be ignored in mean merged agent, hence a higher weight should be assigned to the k-th agent.
>
> >I am confused about how fig 6(a) shows that “CMFQ learns the tactic of besieging, while MFQ tends to confront frontally”.
>
> Thank you for pointing out the unclear statements here. We have revised our manuscript to make the description clearer, and added gifs animators in the supplementary material to facilitate understanding. Please check them in the direction gifs&imgs. In Fig.6(a), several red agents controlled by CMFQ flank the blue agents to besiege them, which is an advanced cooperation tactic. The blue agents controlled by MFQ only tend to be frontal confrontation, while a large number of agents in the rear are in the waiting state.
>
> >It would be clearer if more details and pseudocode of the algorithm CMFQ are added.
>
> It’s a good advice, we have added a pseudocode in Appendix A in the latest manuscript.
> \
> \
> _We sincerely appreciate your comments. Please let us know if you have further feedback._
>
> _Best, Authors_

---

### Author Response · Authors · 2022-11-29
**The collective response to reviewers**

We thank all the reviewers for their time and constructive comments. In response to these comments, we would like to (i) summarize the contributions of our paper, (ii) summarize and clarify the changes made to the paper.

**Summary of the contributions of the paper**

- We analyze the bottleneck of mean-field reinforcement learning (MFRL) in solving the scalability problem. By decomposing the scalability problem into 1) the dimensionality of joint state-action space increases exponentially as agent number increases, and 2) the non-stationarity increases as agent number increases, we could find that MFRL solves the first problem, while the second problem remains largely unresolved. Hence mean-field Q-learning (MFQ) exhibits a strong scalability during training, but a poor scalability during execution. That is, if we increase the number of agents during execution, MFQ will fail rapidly.
- We propose an algorithm named causal mean-field Q-learning (CMFQ) to further alleviate the second problem, thus significantly increases the robustness of MFQ. MFQ characterizes population behavioral information by averaging actions of agents, then obtains an average merged agent which lacks representational ability. CMFQ quantifies the importance degree of each agent by counterfactual inference. Then more reasonable and causality-aware merged agents could be obtained, enabling agents to robustly concentrate on agents that truly matter. Consequently, CMFQ exhibits impressive scalability during both training and execution.
- CMFQ demonstrates a promising and flexible framework for incorporating causal inference into MFRL. The method to calculate causal effects is very flexible. A new algorithm can be obtained by directly replacing the causal module plugin in the framework.

**Summary of the revisions**

Regarding the feedback from reviewers 3dJB, bK4T, and v11b (denoted by R1, R2, and R3 respectively below), we made following changes in the latest manuscript.

- [R1] A pseudcode of CMFQ is added in Appendix A to make our algorithm even clearer.
- [R1, R3] The bound of the Lagrange remainder in Eq.(9) is proved in Appendix B. More explanations for Eq.(9) are added in Section 4.2.
- [R2, R3] Visualization of weights is added in Appendix C, which provides a more in-depth analysis of the reasons why CMFQ is superior to other baselines.
- [R2] New experiment is added in Appendix D to evaluate the applicability of CMFQ.
- [R1, R3] More explanations for Fig.2, Fig.5, and Fig.8 are added in annotations to facilitate understanding.
- [R1] The discussion about Attention-MFQ has been moved to the introduction section.
- [R1] Preliminary of causal inference in section 3.3 is revised for clarity.
- [R1] Some typos are corrected throughout the manuscript.

In addition, the code of the supplemental experiment, the gifs of CMFQ in the battle and predator-prey tasks, and the weight visualizations of CMFQ and Attention-MFQ are added to the supplemental material.

Finally, we would like to thank all the reviewers again for their comments to make our work even stronger. We feel that the points mentioned in the comments have been refined and appended in the revision, and the reviewers' concerns have been addressed in the rebuttal. We invite all reviewers to take another look at our work.

---

### Decision · Program_Chairs · 2023-01-20

**Decision:**

Reject

**Justification For Why Not Higher Score:**

* The story about causality does not hold.  This is a major problem.
* The paper is not well written.  Many sentences have typos and/or grammar issues.

**Justification For Why Not Lower Score:**

NA

**Metareview: Summary, Strengths And Weaknesses:**

The paper proposes a new causal mean field multi-agent RL technique.  Since multiagent systems scale exponentially with the number of agents in the worst case when we take into account all possible interactions, it has become popular recently to consider a mean field approach where interactions are assumed to have an effect only through the mean of all actions.  This paper proposes to relax this mean field assumption by considering a weighted average of the actions instead of a plain average of the actions.

The paper describes the approach by using concepts from causality.  However this is quite confusing since there is no causal model learned and there is no causal inference being performed.  Again the approach boils down to changing the uniform average of actions in mean field theory to a weighted average of actions.  A heuristic is used to set the weights of this heuristic.  It simply measures the distance (KL-divergence) between the policy based on a uniforma average of actions and the policy based on the action of a specific agent.  As pointed out by the reviewers, it is not clear why this heuristic makes sense.  This heusitic assigns higher weights to actions that are far from the average, but why?  This has nothing to do with any notion of causality.  The fact that an action is far from the average of actions does not mean that this action should have a higher degree of causality.  This is not how causality theory would determine that an agent causes some effect on some other agent.

Strength:
* The paper addresses an important weakness of the mean field approach

Weaknesses:
* The story about causality does not hold.  This is a major problem.
* The paper is not well written.  Many sentences have typos and/or grammar issues.